# Tumor-Treating Fields and Related Treatments in the Management of Pediatric Brain Tumors

**DOI:** 10.3390/curroncol32040185

**Published:** 2025-03-21

**Authors:** Julien Rousseau, Sarah Lapointe, David Roberge

**Affiliations:** Centre Hospitalier de l’Université de Montréal, 1051 Sanguinet St., Montreal, QC H2X 3E4, Canada

**Keywords:** pediatric brain tumors, tumor-treating fields, device therapy, high-grade gliomas, pediatric neuro-oncology, central nervous system tumors, treatment approaches

## Abstract

Pediatric primary brain tumors pose significant therapeutic challenges due to their aggressive nature and the critical environment of the developing brain. Traditional modalities like surgery, chemotherapy, and radiotherapy often achieve limited success in high-grade gliomas and embryonal tumors. Tumor-treating fields (TTfields), a non-invasive therapy delivering alternating electric fields, has emerged as a promising approach to disrupt tumor cell division through mechanisms such as mitotic disruption, DNA damage, and tumor microenvironment modulation. TTfields are thought to selectively target dividing tumor cells while sparing healthy, non-dividing cells. While TTfields therapy is FDA-approved for the management of glioblastoma and other cancers, its application in pediatric brain tumors remains under investigation. Preclinical studies reveal its potential in medulloblastoma and ependymoma models, while observational data suggest its safety and feasibility in children. Current research focuses on optimizing TTfields’ efficacy through advanced technologies, including high-intensity arrays, skull remodeling, and integration with immunotherapies such as immune checkpoint inhibitors. Innovative device-based therapies like magnetic field-based technologies further expand the treatment possibilities. As clinical trials progress, TTfields and related modalities offer hope for addressing unmet needs in pediatric neuro-oncology, especially for tumors in challenging locations. Future directions include biomarker identification, tailored protocols, and novel therapeutic combinations to enhance outcomes in pediatric brain tumor management.

## 1. Background

Pediatric brain tumors can present significant treatment challenges due both to their aggressive behavior and to the sensitive environment that is the developing brain. Traditional therapeutics such as surgery, chemotherapy, and radiotherapy have provided limited success in achieving long-term survival, especially for high-grade gliomas and diffuse brainstem tumors. Tumor-treating fields (TTfields), also named alternating electric field therapy, and other magnetic field-based therapies represent novel and promising options for improving outcomes in patients with pediatric brain tumors, especially those for which conventional treatments fall short. The PRISMA guidelines were used to identify publications that were relevant to this literature review.

TTfields, a non-invasive cancer treatment, work by delivering alternating electric fields at low intensity (<2 V/cm) and intermediate frequency (100–300 kHz), which disrupt cancer cell division [1,2]. An electric field (unit, V/m) is a force field generated by charged particles, exerting a force on other charges in its vicinity. A magnetic field (unit, Vs/m^2^ or tesla (T)), on the other hand, is produced by moving electric charges, such as currents, and influences other moving charges. Electric current (unit, ampere (A)) refers to the flow of electric charges through a conductor [3]. Work continues on the mechanism of action of TTfields, but it is generally accepted that they cause polar molecules within dividing tumor cells to misalign during mitosis, ultimately leading to mitotic disruption and cell death (Figure 1). This is accomplished by interfering with microtubule structure and by attracting polarized particles to the cleavage furrow during telophase through a process named “dielectrophoresis” [1,4]. Additional mechanisms by which TTfields exert their antitumoral effects have been described and include induction of autophagy, disruption of DNA damage repair, disruption of cell migration and metastasis, altered cell metabolism, disruption of cell membrane and blood–brain barrier integrity, and modulation of the tumor microenvironment [5]. This specificity for dividing cells of a specific size makes TTfields a targeted treatment for fast-growing tumors, sparing the surrounding healthy tissues, as most somatic cells in the body are non-dividing [6]. This treatment modality can attain deep tumors despite their distance from the arrays [7,8]. Interestingly, cancer cells that divide along the field are more damaged than those that divide perpendicular to the field (thus, the perpendicular positioning of treatment arrays) [7]. Additionally, the combination of TTfields and conventional chemotherapy and/or radiation has shown synergistic effects in preclinical models [9,10,11,12]. An advantage of TTfields over chemotherapy is that it is delivered continuously and can be anatomically focused. The peak efficacy of TTfields appears to be cancer-specific, with smaller cells most impacted by a higher alternating-field frequency and vice versa [6]. This allows for the selection of TTfield frequencies that are specific for each cancer type [13].

Optune, a proprietary device delivering TTfields, is FDA-approved for the management of newly diagnosed and recurrent glioblastoma and malignant pleural mesothelioma and was approved in October 2024 for the management of metastatic non-small cell lung cancer (NSCLC) in combination with chemotherapy [14]. Emerging evidence suggests efficacy in other cancer types. For instance, a phase III trial of patients with locally advance pancreatic adenocarcinoma randomized to gemcitabine and nab-paclitaxel plus TTfields versus gemcitabine and nab-paclitaxel alone, the PANOVA-3 study, recently met its primary outcome and demonstrated improved overall survival (OS) of 16.20 in the former patients vs. 14.16 months in the latter [15]. A recent phase III randomized trial, the METIS study, met its primary endpoint and showed statistically significant improvement in time to progression in patients with brain metastases from NSCLC treated with stereotactic radiosurgery (SRS) plus TTfields versus patients treated with SRS alone, i.e., 21.9 vs. 11.3 months [16]. In the recent INNOVATE-3 trial, TTfields did not prolong OS in patients with platinum-resistant ovarian cancer [17]. The National Comprehensive Cancer Network (NCCN) guidelines list the use of alternating electric field therapy as a Category 1 recommendation for the management of adults with newly diagnosed glioblastoma, as a category 2A recommendation for the management of adults with grade 4 IDH-mutant astrocytoma, and as a Category 2B recommendation for adults with recurrent high-grade glioma (including grade 3–4 IDH-mutant glioma, IDH-wildtype glioblastoma, and H3-mutated high-grade glioma) [18].

The use of TTfields in glioblastoma stems from the publication of phase 2 and 3 clinical trials in recent years. Namely, EF-11 and EF-14 were landmark trials that investigated the use of TTfields in recurrent and newly diagnosed settings, respectively [19,20]. EF-11 showed no statistically significant benefit in progression-free survival (PFS) (2.2 versus 2.1 months) or OS (6.6 versus 6.0 months) when compared to the best standard of care chemotherapy in recurrent glioblastomas. However, survival was comparable in the two groups, and TTfields were better tolerated than chemotherapy [19]. EF-14 demonstrated significant improvements in both PFS (6.7 versus 4.0 months) and OS (20.9 versus 16.0 months) with the addition of TTfields to temozolomide (TMZ) in patients with newly diagnosed glioblastomas [20]. Additionally, a phase 2 trial showed that the combination of TTfields with bevacizumab was safe and feasible in patients with recurrent glioblastoma [21]. Importantly, only patients with supratentorial glioblastoma were included in these trials—standard array configurations were only designed to provide supratentorial electric field coverage [22]. Quality of life was similar or superior in patients undergoing treatment with TTfields compared to those subjected to conventional chemotherapy in EF-11 and was comparable in those treated with TTfields plus TMZ versus those receiving TMZ alone in EF-14 [19,23]. The reason for the difference in the magnitude of survival benefit across different histologies, e.g., glioblastoma versus brain metastases, remains unclear and needs to be further studied.

TTfields are administered via a portable device that is worn by the patients in a continuous fashion. The apparatus is recommended to be used at least 18 h per day. In a post hoc analysis of EF-14, the outcomes appeared to be superior in patients with glioblastoma who wore the device for longer than 18 h per day compared to patients undergoing treatment administrations of shorter durations [24]. Long-term data from the EF-14 study showed that patients who wore the device for more than 90% of the time had a 5-year survival rate of 29.3% [24]. TTfields are administered via transducer arrays placed on the skin. The head must be shaved every 3–4 days during treatment [13]. TTfields delivery is optimized with the help of the FDA-approved software NovoTAL (Novocure, Retrieved from https://www.optunegiohcp.com/about-optune-cancer-treatment/novotal-system on 18 March 2025), which allows for maximizing TTfield delivery at the tumor site [1]. The field generator is powered by a portable battery and can be directly connected to an electrical outlet overnight or whenever the patient is at home. The apparatus can be worn in a backpack or belt bag to facilitate portability. Since the publication of the original landmark trials, a second-generation Optune system was developed, Optune Gio^®^, which is lighter and more easily handled [25]. New lighter and more flexible arrays were recently approved by the FDA [26]. Additionally, outcomes are maximal in patients with higher performance status and treated at first rather than subsequent to recurrence [19,27]. TTfields are a relatively expensive treatment that is typically billed as an all-inclusive monthly service.

TTfields are generally well tolerated, and their side effects can be easily managed. The most common adverse event is skin toxicity, which includes erythema (dermatitis), cutaneous ulcers, and infections. It affects 16–52% of treated patients based on clinical trial and post marketing data [19,20,27,28]. Dermatitis usually occurs within 2–6 weeks of treatment and can be prevented and addressed by frequent array relocation, frequent shaving, and careful scalp hygiene [29]. The presence of dermatitis may warrant the use of treatments such as topical corticosteroids and or topical antibiotics if there is skin breakdown [30]. TTfields are otherwise not associated with any systemic toxicity and are well tolerated [6,28]. Of note, studies have shown that while TTfields can cause localized heating, temperature increases are generally well controlled. Computational models have demonstrated that the maximum temperature in the brain tissue typically remains below 40 °C, with the highest temperatures observed on the scalp under the transducers [31,32]. Important contraindications include implanted medical devices such as pacemakers, a skull defect, or bullet fragment(s) [33]. Additionally, the use of TTfields in patients with intracranial hardware such as ventriculoperitoneal shunts appears safe, although this needs to be investigated further.

The devices from Novocure are currently the only clinically approved electrical field delivery systems. New devices are emerging in an attempt to disrupt brain tumor growth via alternate technologies such as magnetic fields and photosensitizers, as well as intratumoral modular therapy (IMT), as will be briefly discussed below.

## 2. Tumor-Treating Fields in Pediatric Brain Tumors

The literature on the use of TTfields and related technologies for the management of pediatric brain tumors is limited. The landmark clinical trials EF-11 and EF-14 only included patients who were 18 years of age and older [19,20]. Pediatric brain tumors are aggressive entities, some of which frequently progress or recur despite the use of chemotherapy and/or radiation therapy, including diffuse midline glioma (and diffuse pontine glioma), medulloblastoma, and ependymoma. There is a need for the development of novel treatment modalities to address such tumor types. Given their documented efficacy against various tumor types in the adult population, non-invasive nature, and good tolerability, TTfields represent an attractive option, particularly for brain tumors located in critical areas such as the brainstem, where surgical resection is often not feasible [30].

The limited available preclinical evidence supports a role for TTfields in the treatment of pediatric brain tumors. Branter et al. showed that electrical therapy via TTfields and deep brain stimulation negatively impacted cell viability across various primary brain tumor cell lines, including medulloblastoma and ependymoma cell lines, while sparing mature astrocytes [11]. This suggests that the efficacy of electric fields is not limited to glioblastoma and should be evaluated further across other adult and pediatric brain tumor types. Additionally, using a medulloblastoma cell line, another group showed that TTfields slowed tumor growth and induced apoptosis and necrosis [34]. Other electrotherapies are being developed, some of which are being tested in pediatric brain tumor cell lines and models, such as intratumoral modulation therapy (IMT), a technique involving the use of implantable electrodes that deliver low-intensity electric fields directly into the tumor [35].

The limited clinical literature on the use of TTfields for pediatric tumors mostly consists of observational and post-marketing data from a limited number of patients (Table 1). Nonetheless, these data suggest that the use of TTfields in children with brain tumors is safe, feasible, and well tolerated, including in combination with chemotherapy and radiation therapy. Importantly, compliance is satisfactory in most children, and the average daily usage is comparable to that in adults [28,36,37,38,39,40,41]. Based on these studies, children and adolescents were treated off-label with both the first and the second generations of Optune. High-grade gliomas (anaplastic astrocytoma and oligodendroglioma, glioblastoma, diffuse midline glioma, diffuse hemispheric glioma) were the most commonly encountered tumor types. Additional histologies included ependymoma, meningioma, PNET, and pleomorphic xanthoastrocytoma. Most tumors were supratentorial. The infratentorial array layout was used in one case [42]. In the two included post-marketing surveillance studies, 133 patients aged 3–17 were treated with TTfields and tolerated the treatment well [28,37]. The most common adverse event was a skin reaction, which occurred in 36–37% of the patients. In the post-marketing study by Goldman et al., 14% of the patients developed a serious adverse event that was not directly related to the treatment with TTfields. These adverse events included seizures, infection, and cerebral edema [37]. Adherence to treatment was satisfactory in most reported cases, with an average daily use exceeding 75%. Additionally, TTfields were successfully used in combination with systemic therapies such as temozolomide and bevacizumab. In one case series, three patients treated with TTfields also had a programmable shunt, suggesting that TTfields may be safe in patients with surgical hardware. Most patients included in this case series were able to maintain normal activities during treatment, showing that TTfields may be associated with an acceptable level of quality of life [40]. However, the impact of TTfields on pediatric quality of life has not been directly measured and should be addressed with standardized tools in future prospective studies. Gött et al. showed that the device can be tolerated in very young children and reported a case in which a 3-year-old patient with diffuse midline glioma was treated with TTfields. The average monthly use was low initially, at about 35%, but gradually improved over time and reached 75% after about 4 months. The child did not experience any significant toxicity from the treatment [42]. In summary, the current data indicate that children with brain tumors are suitable candidates for TTfields.

The potential toxicity of TTfields in the developing brain remains unknown. Preclinical studies have shown that the effect of TTfields is specific to dividing cancer cells while sparing healthy cells [43]. Ye et al. evaluated the effects of TTfields with different duty cycles on glioma spheroid cells and normal brain organoids. The study found that higher duty cycles of TTfields (75% and 100%) inhibited the proliferation of glioma cells and increased apoptosis. However, in normal brain organoids, higher duty cycles also resulted in a decrease in neural stem cell markers and an increase in glial fibrillary acidic protein expression, indicating potential neurotoxicity and effects on normal brain development [44]. Although the previously cited post-marketing studies did not reveal any TTfield-associated neurotoxicity in the pediatric population, the impact of this treatment modality on cerebral development and myelination needs to be further studied.

## 3. Perspectives

Currently, there are various active clinical trials involving tumor-treating fields and related technologies for the management of primary brain tumors and brain metastases (Table 2) [45]. Most are early-phase trials. and only two include pediatric subjects. NCT03128047 is a phase 1 trial evaluating the safety of TTfields in children and adolescents aged 1–17 with supratentorial high-grade glioma, either in monotherapy or in combination with chemotherapy. NCT03033992 is a phase 1/2 trial assessing the feasibility and safety of TTfields in children and adolescents aged 5–21 with recurrent, progressive, or refractory supratentorial high-grade glioma or supratentorial ependymoma (stratum 1) and the feasibility and efficacy of TTfields concurrent to standard radiation therapy in children and adolescents aged 3–21 with newly diagnosed diffuse intrinsic pontine glioma [45].

An important area of investigation involves the use of TTfields for tumors beyond gliomas. As described above, there is preclinical evidence that electric fields are efficient against various cell lines, including medulloblastoma, and TTfields are clinically approved for other pathologies such as mesothelioma and metastatic non-small cell lung cancer. Trials are underway to assess the potential benefits of TTfields in treating other pediatric CNS malignancies, such as ependymomas (Table 2). The potential for dissemination across the neuraxis should be taken into account when considering TTfields for tumors such as medulloblastoma or ependymoma. Given that TTfields therapy exerts its therapeutic effects through localized electric fields targeting mitotic activity within the tumor bed, its efficacy would likely be limited for disseminated disease. There are some key differences that may have an influence on the efficacy of TTfields when used in the pediatric population, including an inferior skull thickness and the frequent infratentorial tumor localization [46]. The posterior fossa and brainstem remain challenging locations for the treatment with TTfields. These locations were initially excluded from TTfields trials. Novel array placements and new software to model the electrical fields have, however, expanded the use of TTfields to these locations [47,48]. The METIS trial (Table 2) evaluating the impact of TTfields on brain metastasis from NSCLC following SRS was amended to include posterior fossa tumors. In addition to tumor location, next-generation planning software may be able to account for skull thickness.

One promising area of research is the combination of TTfields with immunotherapies such as immune checkpoint inhibitors. TTfields induce immunogenic cell death, a process in which dying cancer cells release danger signals that stimulate the immune system to attack the tumor [5]. Chen et al. demonstrated that TTfields activate the STING and AIM2 inflammasomes in glioblastoma cells, leading to the production of proinflammatory cytokines and type 1 interferons [49]. This opens the door for combining TTfields with immune-modulating agents to bolster the anti-tumor immune response in brain tumors. Voloshin et al. demonstrated that TTfields lead to immune activation both in vitro and in vivo and that combining TTfields to PD-L1 leads to significantly greater tumor volume reduction than either treatment alone in lung and colon cancer murine models [50]. Of note, a recently completed phase 2 study, the 2-THE-TOP trial, showed that the addition of TTfields and pembrolizumab to adjuvant TMZ in 26 patients with newly diagnosed glioblastoma was well tolerated and resulted in significantly longer PFS (12.0 months versus 5.8 months) and OS (24.8 months versus 14.6 months) when compared to case-matched controls from the EF-24 trial [51]. This represents the basis for the recently opened EF-41 trial, a phase 3 study aiming to investigate the OS benefit of combining TTfields with TMZ and pembrolizumab in adult patients with newly diagnosed glioblastoma. Overall, these results demonstrate the potential benefit of combining TTfields with immunotherapy. There are techniques under development seeking to enhance the efficacy of TTfields. Skull remodeling is a newly described surgical technique that consists in the creation of burr holes to enhance the efficacy of TTfields. This technique was deemed safe in a recently completed phase 1 trial and is currently being evaluated in a phase 2 trial (Table 2) [52]. High-intensity arrays are currently being studied in a pilot study, EF-33 [45]. Additionally, novel device therapies are being developed in neuro-oncology. For instance, the use of deep brain stimulation (DBS) electric fields has shown efficacy against adult and pediatric brain tumor cell lines. Branter et al. demonstrated that while TTfields promote G2/M phase accumulation in brain tumor cell lines, DBS electric fields promote G0 phase accumulation [11]. Another interesting avenue is the exploration of magnetic field-based devices, which share similarities with those delivering TTfields but utilize magnetic fields to disrupt cancer cell growth. These magnetic field-based devices include systems for magnetic field therapies and Triplet State technology. While still in early stages of development, magnetic field therapies may provide an additional non-invasive option for targeting pediatric brain tumors. Whereas tumor-treating fields represent sinusoidal orthogonal electrical fields at very specific frequencies and intensities, magnetic field therapy has been studied in a wide range of configurations (including constant fields). Although electrical and magnetic fields are intertwined, they likely have unique mechanisms of action—fixed magnetic fields do not induce electrical fields. The proposed mechanism of action involves the generation of reactive oxygen species (ROS) and the disruption of mitochondrial function. For instance, rotating magnetic fields (RMFs) have been shown to inhibit mitochondrial respiration, promote oxidative stress, and induce the loss of mitochondrial integrity, leading to cancer cell death even in non-dividing phases. Oscillating magnetic fields (OMFs) selectively induce cytotoxic effects in glioblastoma cells by increasing ROS production and activating caspase 3, while sparing normal cells [53,54]. One magnetic field device, the Nativis Voyager, is a portable device that emits ultra-low radio frequency energy that is thought to disrupt mitosis by interacting with specific cellular targets. It was recently studied in a small cohort of patients with recurrent glioblastoma and was proven to be safe and tolerable [55]. By capturing the radiofrequency energy (RFE) of a specific molecule, such as the siRNA of epidermal growth factor receptor (EGFR), and exposing a human glioblastoma cell line to the same RFE, Ulasov et al. were able to successfully knockdown the expression of EGFR and impair cell viability [56]. Another device, Triplet State technology, utilizes low-intensity magnetic fields as a means to enhance the effects of radiation therapy on brain tumors. This principle was demonstrated in vitro and in vivo, where the addition of weak magnetic fields to ionizing radiation led to decreased glioblastoma cell survival and decreased tumor growth [57]. The CYTOTRON is a distinct medical therapeutic device that administers quantum magnetic resonance therapy. It operates by emitting rotating, targeted radio frequencies combined with an instantaneous magnetic field [58]. In summary, there is ongoing research striving to optimize TTfields and to develop different devices using analogous technologies. Albeit promising, these have yet to be studied in the pediatric population.

## 4. Conclusions

The application of TTfields and related technologies represents a paradigm shift in the management of brain tumors, offering a non-invasive approach with the potential to disrupt cancer cell division at the molecular level. Landmark trials such as EF-11 and EF-14 have established the efficacy of TTfields in adults with glioblastoma in recurrent and newly diagnosed settings, respectively. While TTfields are currently FDA-approved for adult brain tumors, including glioblastoma, their potential for pediatric brain tumors remains an area of active investigation.

Pediatric brain tumors, particularly aggressive entities such as diffuse midline glioma and medulloblastoma, represent a significant unmet challenge due to the limitations of surgery, chemotherapy, and radiotherapy, especially for tumors located in critical areas like the brainstem. Preclinical evidence highlights the potential of TTfields to impact cell viability across multiple tumor types, including medulloblastoma and ependymoma, while sparing healthy cells. Moreover, combination therapies with TTfields and treatments such as TMZ, radiation, or immune checkpoint inhibitors, have shown synergistic effects in preclinical and early clinical studies.

Observational studies of TTfields in pediatric populations have suggested adequate safety, feasibility, and adherence to treatment, even in young children. Importantly, adverse events such as skin reactions are generally manageable, and the absence of systemic toxicity makes TTfields an attractive option for children with high-grade gliomas and other CNS tumors.

Innovations such as high-intensity arrays, skull remodeling, deep brain stimulation, and magnetic field-based therapies are expanding the scope of device-based treatments for brain tumors. Additionally, the development of a newer generation of Optune has made the device more palatable for patients. The development of approaches combining TTfields with other modalities such as immunotherapies offers opportunities to enhance the outcomes further. As clinical trials progress, TTfields could become a cornerstone in the multimodal management of brain tumors. Future directions should include the identification of biomarkers of treatment response, the development of protocols tailored to specific pediatric tumor types, and the expansion of research into treatments for posterior fossa tumors and tumors in other challenging locations. Combining TTfields with other radiotherapy modalities such as craniospinal irradiation could also be explored. Given the promising preclinical and observational data, TTfields represent a viable option for children and adolescents with high-grade gliomas, particularly those with supratentorial tumors, as their non-invasive nature and tolerable side effect profile offer significant advantages. However, their application should focus on well-selected patients with strong parental support to ensure adherence to the daily usage requirement and proper management of treatment-related toxicity, such as scalp care, to maximize their therapeutic efficacy.

## Figures and Tables

**Figure 1 curroncol-32-00185-f001:**
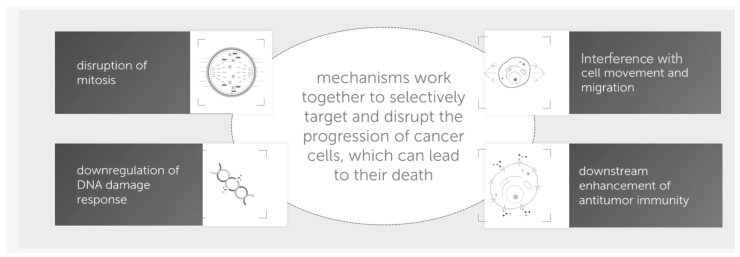
Proposed mechanisms of action of tumor-treating fields (adapted from Novocure^®^ with their permission).

**Table 1 curroncol-32-00185-t001:** Summary of the available clinical evidence for the use of TTfields in patients with pediatric brain tumors.

Authors	Year of Publication	Type of Study	Number of Pediatric Participants (<18 Years Old)	Age	Tumor Type(s)	Tumor Location	Main Findings
Goldman et al. [37]	2022	Post-marketing safety surveillance study	81	3–17	AA (13), AEPN (2), GBM (59), HGG NOS (3), atypical meningioma (1), PNET (1), PXA (1)	Supratentorial (71), infratentorial (4), unknown (6)	Median treatment duration was 81 days; 51 (63%) patients reported at least 1 grade 1 AE; skin reactions were the most common adverse event (36%) and were similar across age groups; 28 SAEs reported, none deemed related to TTfields.
Shi et al. [28]	2020	Post-marketing safety surveillance study	52	3–17	ndGBM (19), rGBM (22), AA/AO (8), others (3)	Unknown	Pediatric participants reported overall less AEs (58%) compared to adult and elderly participants (63–66%). Skin disorders were the most commonly reported AEs (37%).
Crawford et al. [36]	2020	Case series	4	4–16	rHGG	Supratentorial	Treatment duration up to 4 months. No device-related AE; 53–92% compliance rate. All patients died within 2–6 months after TTfield initiation
Green et al. [39]	2017	Case series	3	10–15	GBM (1), HGG (1), DMG (1)	Supratentorial	Patients were treated with TTfields for 5–6 months, until tumor progression. Two patients with partial response, possibly related to RT. Two patients reported no AEs. One patient reported grade 2 skin toxicity with ulceration requiring a 2-day treatment interruption.
Toledano et al. [40] *	2018	Case series	5	11–17	DMG (4), DHG (1)	Supratentorial (1), unknown (4)	Three patients maintained > 90% daily use, one patient with 80% daily use, one patient with 60% daily use. Two patients had disease progression after 86 and 142 days on treatment; treatment duration unknown for the remaining patients. Twelve patient reported minor cutaneous symptoms. Four patients maintained normal activities. Three patients had programmable shunts.
Wölfl et al. [41] *	2017	Case series	3	7–11	AA (1), GBM (2)	Unknown	Duration of use not reported. Average daily use ranged from 71 to 92%. Treatment was well tolerated without significant impact on quality of life. Treatment outcomes not reported.
Gött et al. [42]	2022	Case report	1		DMG	Infratentorial	Infratentorial layout used, in combination with maintenance TMZ. Treatment duration was almost 9 months. Initial daily use was low at 40% but gradually increased to 76% during months 4 to 8. No treatment-related AEs were observed. Repeat brain MRI 1 year after initial biopsy showed response to treatment.
O’Connell et al. [38]	2017	Case report	1	13	rGBM	Supratentorial	Patient maintained ≥ 75% daily use for the entire treatment duration of 13 months. Treatment was well tolerated with minimal scalp irritation. Patient had stable disease for 7 months while administered TTfields.

* abstracts. AA: anaplastic astrocytoma, AEs: adverse events, AEPN: anaplastic ependymoma, AO: anaplastic oligodendroglioma, DHG: diffuse hemispheric glioma, DMG: diffuse midline glioma, GBM: glioblastoma, HGG: high-grade glioma, ndGBM: newly diagnosed glioblastoma, NOS: not otherwise specified, PNET: primitive neuroectodermal tumor, PXA: pleomorphic xanthoastrocytoma, rHGG: recurrent high-grade glioma, rGBM: recurrent glioblastoma, RT: radiation therapy, SAEs: serious adverse events, TMZ: temozolomide, TTfields: tumor-treating fields, US: United States.

**Table 2 curroncol-32-00185-t002:** Active clinical trials involving tumor-treating fields and related technologies in patients with CNS tumors.

Study ID	Study Name	Phase	Country	Population	Pathology	Intervention	Primary Objective/Primary Outcome Measure	**Status**	**Estimated Study Completion**
**ADULTS**									
NCT04471844	TRIDENT	3	International	Adults	Glioblastoma, newly diagnosed	TTfields concomitant to TMZ and RT	Efficacy (OS)	Active, not recruiting	1/2026
NTC06556563	EF-41	3	International	Adults	Glioblastoma, newly diagnosed	TTfields concomitant to TMZ and pembrolizumab	Efficacy (OS)	Recruiting	4/2029
NCT03405792	2-THE-TOP	2	United States	Adults	Glioblastoma, newly diagnosed	TTfields concomitant to TMZ and pembrolizumab	Efficacy (PFS)	Active, not recruiting	9/2025
NCT06140875	BRAIN-RF	1	Germany	Adults	Glioblastoma, newly diagnosed	Combined chemoradiation and radiofrequency electromagnetic field	Efficacy (PFS)	Recruiting	5/2029
NCT03223103	N/A	1	USA	Adults	Glioblastoma, newly diagnosed	TTfields with MTA-based vaccine during adjuvant TMZ	Safety	Active, not recruiting	5/2025
NCT04474353	N/A	1	USA	Adults	Glioblastoma, newly diagnosed	TTfields, combined chemoradiation with SRS, followed by adjuvant TMZ	Safety	Active, not recruiting	11/2025
NCT05086497	N/A	N/A	USA	Adults	Glioblastoma, newly diagnosed	TTfields with advanced array placement based on advanced MRI with spectroscopy	Efficacy (PFS)	Recruiting	6/2026
NCT06558214	OPTIMUS PRIME	2	United States	Adults	Glioblastoma, recurrent/progressive	TTfields, MRI-guided laser ablation, and pembrolizumab	Safety and feasibility	Recruiting	10/2029
NCT04223999	OptimalTTF-2	2	Denmark	Adults	Glioblastoma, recurrent/progressive	skull remodeling plus TTfields plus best practice medical management	Efficacy (12-month OS)	Recruiting	3/2026
NCT04671459	TaRRGET	2	Poland	Adults	Glioblastoma, recurrent/progressive	TTfields and SRS	Efficacy (1-year survival rate)	Active. not recruiting	12/2024
NCT04221503	N/A	2	USA	Adults	Glioblastoma, recurrent/progressive	TTfields and niraparib	Efficacy (CR, PR, or SD)	Active, not recruiting	12/2025
NCT04397679	N/A	1	USA	Adults	Grade 4 glioma, newly diagnosed	partial brain radiation therapy, temozolomide, chloroquine, and tumor-treating fields	Safety (high-grade dermatitis)	Active, not recruiting	4/2026
NCT05310448	N/A	1	USA	Adults	Brainstem glioma	SOC plus TTfields	Safety and tolerability	Recruiting	11/2025
NCT01892397	N/A	2	USA	Adults	Recurrent atypical and malignant meningioma	TTfields monotherapy	Efficacy (PFS)	Active, not recruiting	6/2025
NCT02847559	N/A	2	USA	Adults	Recurrent atypical and malignant meningioma	TTfields plus bevacizumab	Efficacy (PFS-6)	Recruiting	12/2026
NCT02831959	METIS	3	International	Adults	Brain metastases from NSCLC	TTfields following SRS	Time to intracranial progression	Active, not recruiting	12/2024
NCT05341349	N/A	1	USA	Adults	Brain metastases from melanoma	stereotactic radiosurgery and immune checkpoint inhibitors with NovoTTF-100M	Safety	Recruiting	3/2025
**PEDIATRICS**									
NCT03128047	HUMC 1612	1	USA	Pediatrics	Recurrent high-grade glioma and ependymoma	Optune NovoTTF-200A system combined with temozolomide and bevacizumab	Safety and tolerability	Active, not recruiting	6/2024
NCT03033992	PBTC-048	1/2	USA	Pediatrics	Recurrent/progressive/refractory supratentorial high-grade glioma and ependymoma (stratum 1), newly diagnosed DIPG (stratum 2)	Optune NovoTTF-200A system (stratum 1), Optune NovoTTF-200A in combination with RT (stratum 2)	Safety and feasibility (stratum 1), safety, feasibility, and efficacy (stratum 2)	Active, recruiting	9/2032

DIPG: diffuse intrinsic pontine glioma, MTA: mutation-derived tumor antigen, OS: overall survival, PFS: progression-free survival, PFS-6: progression-free survival for 6 months, SOC: standard of care, TTfields: tumor-treating fields, NSCLC: non-small cell lung cancer, TMZ: temozolomide, SRS: stereotactic radiosurgery, USA: United States of America.

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
