# Peer review of "Tumor-Treating Fields and Related Treatments in the Management of Pediatric Brain Tumors"

_curroncol, 2025, doi:10.3390/curroncol32040185_

Round 1
Reviewer 1 Report
Comments and Suggestions for Authors
The authors propose to review the use of tumor treating fields (TTFields) for the management of pediatric brain tumors. This is an important topic that has only been infrequently and incompletely addressed so far and fully deserve a thorough review. The review is well written and easily understandable. A few points could however help improve its quality.
Main concerns:
- It remains unclear whether the authors have used the PRISMA guidelines to identify relevant publications. If yes, it would be important to state this specifically
- Whereas the authors clearly highlight the lack of prospective studies and the fact that most information in pediatric neuro-oncology relies on case reports or small series. The review of those cases only covers a small paragraph (lines 155 to 171). I would strongly suggest to expand this part of the review and include a table to highlight the main characteristics of the publications along with the main outcomes.
- Concerns with TTFields specific to the pediatric population are only marginally addressed: it would be important to develop a specific chapter on this topic which should include (but not limited to) concern of growing and developing brain tissue under TTFields, issues with compliance in the pediatric population (one would expect that the requirement of carrying the Optune device would severely limit the quality of life of children that want to move around freely as much as possible)
- The authors highlight both brainstem gliomas and medulloblastoma as specific targets of interest for TTFields in the pediatric population. These 2 conditions raise specific concerns that are worth expanding in this review: For brainstem gliomas, the ability to adequately deliver 2 fields of TTFields are complicated by the anatomy (at least compared to supratentorial tumors) and medulloblastoma has a high chance of being not a localized but disease that presents metastases at distance. This raises challenges for a typically localized treatment such as TTFields. These points should be discussed by the authors.
Minor concerns:
- In the introduction, it might be worth mentioning that in platinum resistant ovarian cancer TTFields missed its OS endpoint.
- Line 69: please add after (NSCLC): “in combination with chemotherapy”, as TTFields were never used alone.
- Line 73: delete: improved in "improved an improved overall" ….
- Line 103-104: the link between compliance and extended survival was not a prespecified analysis and subject to quite significant bias. This sentence should be rewritten.
Author Response
Reviewer 1
Main concerns:
Comment 1: It remains unclear whether the authors have used the PRISMA guidelines to identify relevant publications.
Answer 1: Yes, the PRISMA guidelines were used to identify relevant publications, which we have clarified in the text on lines 36-37.
Comment 2: Whereas the authors clearly highlight the lack of prospective studies and the fact that most information in pediatric neuro-oncology relies on case reports or small series. The review of those cases only covers a small paragraph (lines 155 to 171).I would strongly suggest to expand this part of the review and include a table to highlight the main characteristics of the publications along with the main outcomes.
Answer 2: Thank you for the suggestion. We have added a new table, Table 1 (active clinical trials can now be found in Table 2), which summarizes the available evidence. We have also expanded the paragraph to reflect in more details the cited literature (lines 171-190)
Comment 3: Concerns with TTFields specific to the pediatric population are only marginally addressed: it would be important to develop a specific chapter on this topic which should include (but not limited to) 1) concern of growing and developing brain tissue under TTFields, 2) issues with compliance in the pediatric population (one would expect that the requirement of carrying the Optune device would severely limit the quality of life of children that want to move around freely as much as possible)
Answer 3: Thank you for this recommendation. We added a paragraph on the potential toxicity of TTF on the developing brain (lines 197-207) We also addressed the lack of evidence regarding the impact of TTF on quality of life (lines 186-190).
Comment 4: The authors highlight both brainstem gliomas and medulloblastoma as specific targets of interest for TTFields in the pediatric population. These 2 conditions raise specific concerns that are worth expanding in this review: 1) For brainstem gliomas, the ability to adequately deliver 2 fields of TTFields are complicated by the anatomy (at least compared to supratentorial tumors) and 2) medulloblastoma has a high chance of being not a localized but disease that presents metastases at distance. This raises challenges for a typically localized treatment such as TTFields. These points should be discussed by the authors.
Answer 4: Thank you for the suggestion. We have expanded the Perspectives section to address the question of drop metastases (lines 224-228) and the question of TTF in posterior fossa and brainstem tumors (lines 231-235).
Minor concerns:
Comment 5: In the introduction, it might be worth mentioning that in platinum resistant ovarian cancer TTFields missed its OS endpoint.
Answer 5: We added this information in the Background section (line 79-80)
Comment 6: Line 69: please add after (NSCLC): “in combination with chemotherapy”, as TTFields were never used alone.
Answer 6: Thank you for the clarification. We have adjusted the sentence accordingly (lines 70-71).
Comment 7: Line 73: delete: improved in "improved an improved overall" ….
Answer 7: we have made the correction on line 75.
Comment 8: Line 103-104: the link between compliance and extended survival was not a prespecified analysis and subject to quite significant bias. This sentence should be rewritten.
Answer 8: thank you for the clarification. We have rewritten the sentence and mentioned that this was a post hoc analysis (lines 106-109)
Reviewer 2 Report
Comments and Suggestions for Authors
Introduction
The authors have written an article which gives an overview of the current clinical progress on the application of Tumor Treating Fields (TTF). They begin by providing a background on the characteristics of TTF therapy and the mechanism of action that it is based upon. They then do a survey of the clinical applications of TTF on adult-derived gliomas and then a very brief parallel survey on TTF theapies against pediatric glioblastomas and ependymomas. They end their review article by providing a perspective on future directions of TTF therapy, particularly as it applies to pediatric brain tumors and the combination of TTF with related technologies for the management of brain tumors. TTF is an up and coming therapy for many types of cancers including brain cancer so a review article is timely. However, some issues should be addressed before the article is suitable for publication.
Key Critiques
Although the title of the review article (“Tumor-Treating Fields and Related Treatments in the Management of Pediatric Brain Tumors”) suggests that the focus is on pediatric tumors, the majority of the article’s descriptions in fact go towards adult-derived glioblastomas. Even those sections of the review that are specifically devoted to pediatric tumor give significant expositions to adult-derived brain cancers. We thus suggest that the title of the review article should be altered to: “Tumor-Treating Fields and Related Treatments in the Management of Adult-derived and Pediatric Brain Tumors”.
In addition, it is not certain if the authors have looked into all types of pediatric tumors and the treatment by TTF thereof. It is not certain if the authors have considered the treatment of Diffuse Invasive Pontine Gliomas (DIPG) by TTF. DIPG is a major pediatric brain tumor and any report on TTF-related therapy on DIPG, in our opinion, should be included within the author’s review article. Since much of the review devotes itself to adult-derived glioblastomas, the authors would do well to provide also, comparisons and contrasts of TTF treatment on adult versus pediatric tumors. What are the similarities and what are the differences in protocols between adult-focused versus pediatric-focused regimens?
Finally, while in the future perspective section, the authors do describe the potential of using magnetic fields as a therapeutic against cancers, it is not clear to the reviewers what should be the mechanism of action. Do magnetic fields work by producing an electric field orthogonal to the direction of the magnetic field and act
Mthen as de facto TTF or is there a direct effect by the magnetic field? Are the sites of action the same as those of TTFs’? Do magnetic fields act upon the mitotic spindle of dividing cells or are there other sites of action? The authors should try to include reports or data that could address these questions.
Minor Critiques:
Page 1, line 37, the line should read “TTF, a non-invasive cancer treatment, works by delivering alternating electric fields……”
Page 2, Lines 55-57 – perhaps after this sentence, one should mention why TTF would target primarily dividing cancerous cells, i.e. that most normal, somatic cells in the body are non-dividing. Hence they do not form the mitotic spindle which contain the dipole moments that TTF would act upon.
Page 7, Lines 226-238 – Exposition on the use of low frequency magnetic fields to treat brain tumors. If this anti-cancer therapy is valid then what are the implications of clinical grade MRIs and their applicability of being anti-cancer therapies? Has anyone considered this?
Author Response
Reviewer 2
Key Critiques
Comment 1: Although the title of the review article (“Tumor-Treating Fields and Related Treatments in the Management of Pediatric Brain Tumors”) suggests that the focus is on pediatric tumors, the majority of the article’s descriptions in fact go towards adult-derived glioblastomas. Even those sections of the review that are specifically devoted to pediatric tumor give significant expositions to adult-derived brain cancers. We thus suggest that the title of the review article should be altered to: “Tumor-Treating Fields and Related Treatments in the Management of Adult-derived and Pediatric Brain Tumors”.
Answer 1: Thank you for the comment. The purpose of this review was to review the available literature on TTF in pediatric neuro-oncology. We agree with the reviewer that one has to rely on adult data when considering pediatric TTF use. The special issue for which this review was written focuses on pediatric neuro-oncology. There already exist more comprehensive reviews on TTF in adult neuro-oncology and, for this reason, we would prefer not changing the title.
Comment 2: In addition, it is not certain if the authors have looked into all types of pediatric tumors and the treatment by TTF thereof. It is not certain if the authors have considered the treatment of Diffuse Invasive Pontine Gliomas (DIPG) by TTF. DIPG is a major pediatric brain tumor and any report on TTF-related therapy on DIPG, in our opinion, should be included within the author’s review article. Since much of the review devotes itself to adult-derived glioblastomas, the authors would do well to provide also, comparisons and contrasts of TTF treatment on adult versus pediatric tumors. What are the similarities and what are the differences in protocols between adult-focused versus pediatric-focused regimens?
Answer 2: The available clinical evidence for the use of TTF in all pediatric brain tumors has been reviewed, including DMG/DIPG. As suggested above, we have expanded the section describing the clinical evidence (lines 171-190) and have summarized the literature in a new table, Table 1. There is still only one TTF device which has achieved regulatory approval (other devices with different frequency or transducers are at different levels of clinical trials) — differences in pediatrics would be related to transducer placement (more posterior fossa/brainstem tumors), tolerance and integration with other therapies. It remains that TTF is typically used for as many hours possible each day and for long periods. We had expanded the Perspectives section to address some of the challenges related to the administration of TTF in pediatric brain tumors (lines 224-236).
Comment 3: Finally, while in the future perspective section, the authors do describe the potential of using magnetic fields as a therapeutic against cancers, it is not clear to the reviewers what should be the mechanism of action. 1) Do magnetic fields work by producing an electric field orthogonal to the direction of the magnetic field and act as de facto TTF or is there a direct effect by the magnetic field? 2) Are the sites of action the same as those of TTFs’? Do magnetic fields act upon the mitotic spindle of dividing cells or are there other sites of action? 3) The authors should try to include reports or data that could address these questions.
Answer 3: Tumor treating fields represent sinusoidal orthogonal electrical fields at very specific frequencies and intensity. Magnetic field therapy has been studied in a wide range of configurations (including constant fields). Although both electrical and magnetic fields are intertwined, they likely have unique mechanisms of action — fixed magnetic fields do not induce electrical fields. Magnetic fields have unique proposed mechanisms of action, which we have clarified in lines 301-311.
Minor Critiques:
Comment 4: Page 1, line 37, the line should read “TTF, a non-invasive cancer treatment, works by delivering alternating electric fields……”
Answer 4: We have made the correction on line 38. Thank you.
Comment 5: Page 2, Lines 55-57 – perhaps after this sentence, one should mention why TTF would target primarily dividing cancerous cells, i.e. that most normal, somatic cells in the body are non-dividing. Hence they do not form the mitotic spindle which contain the dipole moments that TTF would act upon.
Answer 5: Thank you for the suggestion. We have modified the sentence on lines 53-55 to reflect this suggestion.
Comment 6: Page 7, Lines 226-238 – Exposition on the use of low frequency magnetic fields to treat brain tumors. If this anti-cancer therapy is valid then what are the implications of clinical grade MRIs and their applicability of being anti-cancer therapies? Has anyone considered this?
Answer 6: The magnetic fields used in MRI are static, high-frequency and of very short duration — which differ from the low-frequency oscillating magnetic fields used in LF-MF therapy. Therefore, MRI machines are not currently applicable as anti-cancer therapies in the same way as LF-MFs.
Reviewer 3 Report
Comments and Suggestions for Authors
This is a well written and clearly explained overview of the potential pediatric use of TTF, approved for adults. Some suggestions for the authors to consider:
Line 129. TTF delivers an EMF, not an electrical field, as the authors point out elsewhere in the paper.
Line 144. I would say the primary rationale for TTF in peds is that it works [somewhat] in adults.
Line 171. I might have missed something, but mentioning data on effectiveness of TTF is partly dependent on total time used, more time = better results.
Line 184 and throughout the ms., I think we should not be using the term “electrical field” re. TTF device. The current is oscillating and that equals an EMF, not an electrical field.
Line 210. Please list the OS from the study ref 45.
Line 223 Again please differentiate “magnetic field” from EMF. Simple permanent magnets produce a magnetic field and I don’t think that is being explored. Again the magnetic devices are creating an EMF aren’t they ?
Line 226. Tripletstate technology must have a diagram and an explanation of the technology. Probably this requires several paragraphs. It is complicated and very few of your readers will understand that technology.
Some minor points:
Line 44. May be better to say “TTF” to keep consistency of terms.
Since TTF uses alternating current should we not be calling the resulting field an EMF ?
Line 59. Did the authors want to include preclinical TTF studies using combinations of repurposed drugs as augmentation maneuvre ?
Line 67. Would it not be correct to say that ä proprietary device to deliver TTF was…” not the TTF itself ?
Line 77. Would the authors like to comment on the considerably stronger effect of TTF on NSCLC mets compared to the welcome, but weak, effect on gliomas ?
Line 127. Would the authors like to mention the brain tissue heating effect of TTF here ?
Author Response
Comment 1: Line 129. TTF delivers an EMF, not an electrical field, as the authors point out elsewhere in the paper.
Answer 1: Tumor treating fields (TTFields) deliver electric fields. TTFields are low-intensity (1-3 V/cm), intermediate-frequency (100-500 kHz) alternating electric fields that are applied to the tumor region via transducer arrays placed on the skin. The difference between electromagnetic fields and electric fields lies in their composition and propagation. Electric fields are generated by electric charges and can exist independently. They exert force on other electric charges within the field. Electromagnetic fields, on the other hand, are composed of both electric and magnetic fields that oscillate perpendicularly to each other and to the direction of wave propagation. Electromagnetic fields are typically associated with the propagation of electromagnetic waves, such as light or radio waves.
Comment 2: Line 144. I would say the primary rationale for TTF in peds is that it works [somewhat] in adults.
Answer 2: Thank you for your comment. We have added this argument to the text (lines 150-151).
Comment 3: Line 171. I might have missed something, but mentioning data on effectiveness of TTF is partly dependent on total time used, more time = better results.
Answer 3: Thank you for your comment. We agree that increase daily use is associated with better outcomes, as is outlined on lines 106-109.
Comment 4: Line 184 and throughout the ms., I think we should not be using the term “electrical field” re. TTF device. The current is oscillating and that equals an EMF, not an electrical field.
Answer 4: Please see answer to the first comment.
Comment 5: Line 210. Please list the OS from the study ref 45.
Answer 5: We added the PFS and OS values on lines 281-282.
Comment 6: Line 223 Again please differentiate “magnetic field” from EMF. Simple permanent magnets produce a magnetic field and I don’t think that is being explored. Again the magnetic devices are creating an EMF aren’t they ?
Answer 6: Thank you for this comment. Please see the definitions for electrical and magnetic fields on lines 41-43. We have also added a section that further distinguishes magnetic from electrical on lines 301-311.
Comment 7: Line 226. Tripletstate technology must have a diagram and an explanation of the technology. Probably this requires several paragraphs. It is complicated and very few of your readers will understand that technology.
Answer 7: Thank for this suggestion. The aim of that section of the paper (Perspectives) is to briefly introduce recently developed devices that use magnetic fields and to provide references for readers who would like to explore this topic further. For that reason, we will not expand the paragraph or provide additional diagrams.
Some minor points:
Comment 8: Line 44. May be better to say “TTF” to keep consistency of terms.
Since TTF uses alternating current should we not be calling the resulting field an EMF ?
Answer 8: Thank you for the commend. We have replaced “alternating electrical fields” by “TTF” on line 45. Regarding the question of EMF, please refer to our answer to your first comment.
Comment 9: Line 59. Did the authors want to include preclinical TTF studies using combinations of repurposed drugs as augmentation maneuvre ?
Answer 9: Thank you for the comment. The studies cited on line 59 describe preclinical experiments showing the synergistic effect of TTF when combined with chemotherapy agents.
Comment 10: Line 67. Would it not be correct to say that ä proprietary device to deliver TTF was…” not the TTF itself ?
Answer 10: Thank you for the suggestion. We have made the correction on lines 68-70.
Comment 11: Line 77. Would the authors like to comment on the considerably stronger effect of TTF on NSCLC mets compared to the welcome, but weak, effect on gliomas ?
Answer 11: Thank you for the suggestion, we have added a sentence addressing this question at the end of the paragraph, on lines 102-104.
Comment 12: Line 127. Would the authors like to mention the brain tissue heating effect of TTF here ?
Answer 12: Thank you for the recommendation. Lines 131-134 include a new section on the heating effect of TTF.
Round 2
Reviewer 1 Report
Comments and Suggestions for Authors
The revised version of the manuscript is considerably improved from the first version and, in my opinion, acceptable for publication.
2 Minor comments:
- I would suggest to change the abbreviation TTF to TTfields as TTF is generally understood to stand for time to treatment failure
- I am not sure that the column Country is necessary or provides important information- This column could be erased, which would provide additional space for the column "main findings"
Author Response
2 Minor comments:
COMMENT 1 : I would suggest to change the abbreviation TTF to TTfields as TTF is generally understood to stand for time to treatment failure
ANSWER 1: we have made the modification in the manuscript.
COMMENT 2: I am not sure that the column Country is necessary or provides important information- This column could be erased, which would provide additional space for the column "main findings"
ANSWER 2: We have removed the column.